# LVM-Lite: Training Large Vision Models with Efficient Sequential Modeling

## Abstract

Generative pre-training has significantly advanced natural language understanding. Building upon this success, recent research begins to innovate Large Vision Models (LVM) by leveraging large-scale pre-training on visual sequences, where simultaneous consideration of image token sequences within single images and across a set of images is of key importance. This paper shows that sequential modeling on single images and across multiple images can be efficiently and effectively decoupled. We introduce a two-stage learning pipeline, starting with single-image pre-training, followed by fine-tuning on long image/video sequences. We term this method Large Vision Model Lite (LVM-Lite). Extensive experiments showcase the impressive performance of LVM-Lite across various generative and discriminative benchmarks, comparable to specifically trained models without the need for task-specific training. Importantly, LVM-Lite accelerates training speed substantially up to 2.7× and demonstrates strong scalability.

## 1 Introduction

The scaling of models in both natural language processing (NLP) (Devlin et al., 2018; Chowdhery et al., 2022; Touvron et al., 2023) and computer vision (Radford et al., 2021; He et al., 2021; Dosovitskiy et al., 2020) has led to significant advancements. In NLP, large language models (LLMs) like GPT (Radford et al., 2019; Brown et al., 2020; Achiam et al., 2023) have revolutionized the field, demonstrating the power of pre-trained models in understanding and generating text through in-context learning. Similarly, in computer vision, scalable methods such as CLIP (Radford et al., 2021), masked image modeling (He et al., 2021; Bao et al., 2021; Xie et al., 2021), and diffusion models (Dhariwal & Nichol, 2021; Rombach et al., 2021) have pushed the boundaries of image understanding and synthesis. Recently, the application of autoregressive pre-training strategies in vision (Yu et al., 2021b; Esser et al., 2021a; Yu et al., 2021a; 2022; El-Nouby et al., 2024; Bai et al., 2023; Ren et al., 2023), inspired by the success of LLMs, has shown promising progress. This scaling up across domains underscores models' ongoing evolution and potential to address increasingly complex tasks by learning from vast amounts of data.

Nonetheless, the remarkable performance of these advanced models often comes with high computational costs, limiting access for researchers without substantial computational resources. For instance, training the LLaMA model (Touvron et al., 2023) requires up to 2,000 GPUs and 1.7 million GPU hours. This paper focuses on efficient sequential modeling in the context of the recently developed Large Vision Models (LVM) (Bai et al., 2023), which shows impressive scalability and proficiency in in-context learning for visual tasks. The success of LVM largely stems from the scale of the model and more crucially the extensive scale of data, encompassing a vast array of random image sequences, video sequences, and image/video-annotation pairs. Notably, random image sequences, comprising 90% of the total dataset tokens and constructed by concatenating single, weakly related or even unrelated image tokens, consume the most computation. Due to the quadratic complexity of self-attention *w.r.t.* token length, training these random images as a unified sequence incurs higher computational costs compared to training them as independent single images. Additionally, these random image sequences, unlike natural language sequences, may harbor higher noise for next-token prediction.

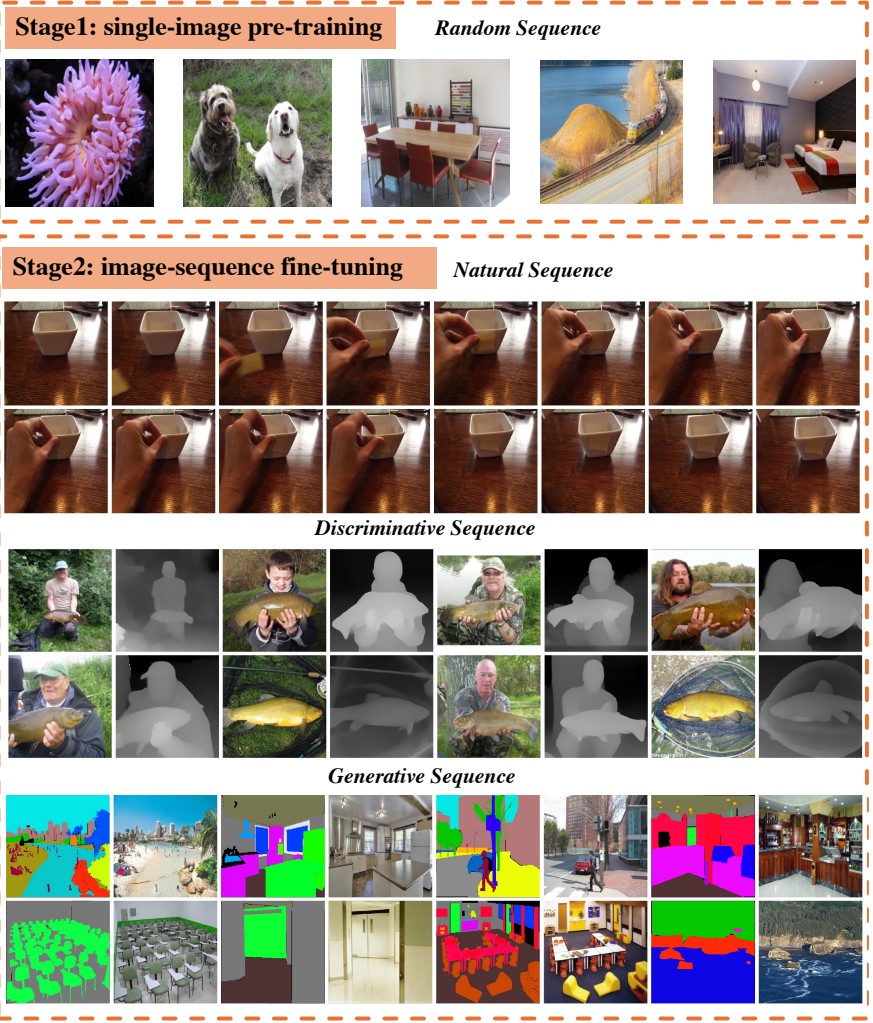

Figure 1: **Overview of the proposed LVM-Lite**. Training process begins with the use of single-image tokens with a reduced token length, followed by short fine-tuning on meaningful visual sequences. Once trained, LVM-Lite can adapt to various vision tasks through in-context generation.

Motivated by these observations, we develop a two-stage training pipeline for efficient and scalable visual sequential modeling. Illustrated in Figure 1, our approach begins with dedicated pre-training on single images, followed by fine-tuning on carefully curated long image/video sequences. This decoupling allows us to efficiently scale up the model in pre-training, further enabling flexible task adaptation in fine-tuning. We name this framework as Large Vision Model Lite (LVM-Lite). Comprehensive experiments showcase its high training efficiency and strong in-context generation capabilities, *e.g.*, we can attain up to 2.7× training speed gains without compromising performance. Indeed, LVM-Lite achieves comparable performance to specifically-trained models across various generative and discriminative benchmarks, including video generation, image generation, and image understanding, without the need for task-specific training.

## 2 Large Vision Model Lite

We first revisit LVM (Bai et al., 2023) in Sec. 2.1. Next, we present a two-stage training framework, encompassing single-image pre-training followed by sequence fine-tuning to speed up LVM training.

### 2.1 A Close Look at LVM Training

**Pre-tokenization and training.** LVM utilizes a VQ-GAN model(Esser et al., 2021b), comprising an encoder $E$ to map input images into a latent space and a decoder $G$ for reconstruction from the latent representation. LVM pre-tokenizes each image $x \in \mathbb{R}^{H \times W \times C}$ into a latent representation $\hat{z} = E(x)$, where $\hat{z} \in \mathbb{R}^{H' \times W' \times D}$, $H' = \frac{H}{16}$ and $W' = \frac{W}{16}$. Subsequently, the quantization module $q$ quantizes each spatial code $\hat{z}_{ij}$ at position $(i,j)$ in the feature map to its nearest codebook entry to form the tokens $z_q$ using the following formula:

$$z_q = q(\hat{z}) = \left( \underset{z_k \in \mathcal{Z}}{\mathrm{argmin}} \, \|\hat{z}_{ij} - z_k\| \right)$$

where $k$ indexes entries within the codebook $\mathcal{Z}$. We flatten $z_q \in \mathbb{R}^{H' \times W'}$ into 1D dimension, tokenizing each image into $S = H' \times W'$ tokens. Thus a visual sequence comprising $N$ images has a total sequence length of $L = N \times S$, denoted as $\mathbf{Z} = \{z_{q_1}, z_{q_2}, ..., z_{q_L}\}$. A decoder-only transformer is trained to predict each token in $\mathbf{Z}$ given all preceding tokens, with an autoregressive loss:

$$\mathcal{L} = \sum_{m=1}^{L-1} \log p(z_{q_{m+1}} | z_{q_{1:m}})$$

which considers the entire image sequence $\mathbf{Z}$, rather than individual images or smaller segments.

**Generation.** After training, the model generates new tokens by sampling from the probability distribution of the next token, modulated by a temperature parameter $T$. Additionally, we employ a top-$K$ sampling strategy to restrict the selection of the next token to the $K$ most probable options predicted by the model. The predicted token sequence $\mathbf{Z_g} = \{z_{g_1}, z_{g_2}, ..., z_{g_n}\}$ is then passed through VQ-GAN's decoder $G$ to reconstruct the image.

$$\hat{x} = G(\mathbf{Z_g})$$

The length $n$ of generated tokens $\mathbf{Z_g}$ controls the number of generated images.

**Training configuration.** Following the setup in previous work (Chang et al., 2022), the VQ-GAN model is trained on LAION-2B (Schuhmann et al., 2022) with a codebook of 8192 entries. Our decoder-only transformer is based on the LLaMA model (Touvron et al., 2023). Each image is represented as a sequence of $S = 256$ discrete tokens. A visual sequence is constructed from $N = 16$ randomly sampled images for training. This forms our standard training protocol with a total sequence length of $L = 4096$.

### 2.2 Single-image Pre-training

From a detailed profiling of LVM (Bai et al., 2023) training, most training efforts are spent on single-image datasets like LAION, where 16 random images form a visual sequence. These random sequences take a substantial amount of compute and carry more noise compared to natural visual sequences (*e.g.*, videos). Therefore, we explore the feasibility of training on individual images from single-image datasets with reduced context length $L$, followed by fine-tuning on curated or natural image sequences.

Specifically, we reduce the number of images from $N = 16$ to $N = 1$ and the sequence length from $L = 4096$ to $L = 256$ in the single-image pre-training stage. With the same training objective, the model now predicts the next token using only preceding tokens from the same image. This strategy significantly decreases model FLOPs, reducing attention complexity from $\mathrm{O}(N^2 S^2)$ to $\mathrm{O}(S^2)$. For instance, the model's FLOPs in our largest 3B model can be reduced by a substantial factor ($19.4\times$, from 33 to 1.7 TFLOPs).

In practice, we keep the total number of training tokens the same as in LVM. We evaluate the effectiveness of this two-stage approach with model performance on downstream tasks after second-stage fine-tuning. As detailed in Table 4, this approach largely accelerates training with comparable performance. For instance, our largest 3B model is $\sim 2.1\times$ faster than our baseline (62k *vs.*130k in TPU-v3 core hours). Next, we delve into the specifics of the second-stage fine-tuning.

### 2.3 Image-Sequence Fine-tuning

LVM (Bai et al., 2023) establishes the significance of sequence data from videos, images with annotations, and videos with annotations. We curate our sequence dataset similarly into three categories for fine-tuning (also visually shown in the bottom left of Figure 1):

- *Natural Sequence:* This category primarily comprises video frames and diverse existing datasets (Soomro et al., 2012; Perazzi et al., 2016; Kuehne et al., 2011; Monfort et al., 2019; 2021; Reizenstein et al., 2021; Goyal et al., 2017b; Das et al., 2013; Carreira et al., 2019; Materzynska et al., 2019; Li et al., 2021; Sigurdsson et al., 2018; Murray et al., 2012; Grauman et al., 2022), along with 3D image data (Deitke et al., 2023). We generate sequences of 16 frames by random sampling.

- *Discriminative Sequence:* We utilize human-labeled image datasets such as Cityscapes(Cordts et al., 2016), ADE20K(Zhou et al., 2017), COCO(Lin et al., 2014), and video datasets like Co3D(Reizenstein et al., 2021), ViPSeg(Miao et al., 2022), VOS(Xu et al., 2018a). Following LVM (Bai et al., 2023), we also employ models (Soria et al., 2020; Cheng et al., 2022; Yang et al., 2024) to generate pseudo labels on ImageNet-1K(Deng et al., 2009). Discriminative sequences are formed by repeated sampling of "image, annotation" pairs eight times to align with the context length.

- *Generative Sequence:* We invert the order of pairs in the discriminative sequence to "annotation, image" and repeat sampling eight times to construct generative sequences.

### 2.4 In-context Evaluation

LVM-Lite performs in-context generation by structuring the input sequence as "prompt, query". The prompts consist of images of questions and answers to allude to the task (*e.g.*, "image, segmentation map" for the semantic segmentation task). The model then generates contents based on the query image. We elaborate on our evaluation designs for various visual benchmarks.

For video generation, the visual prompt consists of the first five frames sampled from a video, adhering to the common practice in video prediction tasks (Yu et al., 2023; Skorokhodov et al., 2021). We then prompt the model to generate the subsequent 11 frames, sampling one clip from each video for evaluation. For image generation, we use "annotation, image" pair to specify the task, such as "segmentation map, image" for segmentation-to-image generation. Similar inputs apply to edge-map-to-image and depth-map-to-image generation. For image understanding, we evaluate on widely recognized semantic segmentation benchmarks with "image, segmentation map" prompt. The segmentation map is generated by applying KNN within a predefined color map as described in (Wang et al., 2023a) aligned with official sources (Zhou et al., 2017; Cordts et al., 2016).

## 3 Experiments

### 3.1 Main Results

We demonstrate the effectiveness of LVM-Lite on various generative and discriminative benchmarks, including video generation, image generation, and image understanding.

**Data.** Following LVM (Bai et al., 2023), we utilize large-scale DataComp-1B (Gadre et al., 2023) dataset for single-image pre-training and three types of visual sequences for fine-tuning, including natural visual sequences (*e.g.*, videos and 3D images), discriminative sequences and generative sequences (see details in Appendix). We preprocess single-image data into 256 tokens and visual sequence data (16 frames or 8 image-annotation pairs) into 4096 tokens with a pre-trained tokenizer and train the model once on all data.

**Training and Evaluation.** Following LVM (Bai et al., 2023)'s computation setting, we pre-train our model on ∼300B tokens from images in Datacomp-1B; this is equivalent to ∼900 training epochs on ImageNet-1K (Deng et al., 2009). The fine-tuning phase uses 96B tokens (∼300 ImageNet-1K epochs). Batch sizes of

Table 1: **Comparison on video generation.** We generated videos at $16 \times 256 \times 256$ resolution given the first 5-frames as prompts following (Yu et al., 2023) and computed FVD with ground truth. *The UCF-101 results are class-conditional (Hong et al., 2023; Yu et al., 2023).

| Model | UCF-101 | | SS-V2 | K600 |
|---|---|---|---|---|
| | FVD-16f ↓ | IS ↑ | FVD-16f ↓ | FVD-16f ↓ |
| CogVideo*(Hong et al., 2023) | 626.0 | 50.5 | - | 109.2 |
| MAGVIT*(Yu et al., 2023) | 76.0 | 89.3 | 31.4 | 9.9 |
| LVM-lite-300M | 252.2 | 33.7 | 172.1 | 193.1 |
| LVM-lite-1B | 222.7 | **34.9** | 165.3 | 180.9 |
| LVM-lite-3B | **188.2** | 33.8 | **138.9** | **166.3** |
| Ground truth rec. | 119.0 | 35.9 | 84.1 | 98.6 |

Table 2: **Comparison on conditional image generation.** Images resolution is set to $256 \times 256$. Seg.: segmentation mask. Depth: relative depth map. Edge: edge map. N/A: not applicable.

| Model | ImageNet-1K | | | | | | ADE20K-G | City.-G |
|---|---|---|---|---|---|---|---|---|
| | FID ↓ | | | IS ↑ | | | FID ↓ | FID ↓ |
| Pix2PixHD(Wang et al., 2018) | N/A | | | | | | 73.3 | 104.7 |
| DP-SIMS(Berrada et al., 2023) | N/A | | | | | | 22.7 | 38.2 |
| VQ-GAN(Esser et al., 2021b) | 5.2 | | | 280.3 | | | 33.5 | N/A |
| synthesis condition | seg. | | depth | | edge | | seg. | |
| LVM-lite-300M | 70.9 | 13.0 | 56.8 | 17.5 | 56.7 | 20.7 | 41.2 | 92.9 |
| LVM-lite-1B | 57.1 | 17.1 | 43.3 | 24.2 | 43.5 | 27.1 | **42.4** | 85.3 |
| LVM-lite-3B | **42.3** | **24.1** | **31.9** | **33.7** | **33.7** | **35.6** | 39.7 | 84.1 |
| Ground truth rec. | 8.0 | | | 338.0 | | | 20.4 | 72.2 |

8192 and 512 are used for context lengths of 256 and 4096, respectively. We use the AdamW optimizer with a base learning rate of 1.5e-4, tapering to an end learning rate of 1.5e-5. The learning rate for fine-tuning starts at 1.5e-5 and ends at 1.5e-6.

We conduct in-context evaluation following the settings in Section 2.4. Specifically, for video generation capabilities, we evaluate on UCF-101 (Soomro et al., 2012), Something-Something V2 (SS-V2) (Goyal et al., 2017b) and Kinetics-600 (K600) (Carreira et al., 2018). Frame prediction performance is measured with the 16-frame FVD(Unterthiner et al., 2019) and IS metric, leveraging a C3D model (Tran et al., 2015) on UCF-101. For image generation, we evaluate conditional image generation on ImageNet-1K (Deng et al., 2009) and ADE-20K (Zhou et al., 2017). Performance is assessed using FID (Parmar et al., 2022) and IS (Salimans et al., 2016) following the ADM protocol (Dhariwal & Nichol, 2021). Prompts and queries include relative depth maps from depthanything (Yang et al., 2024), segmentation maps from mask2former (Cheng et al., 2022), and edge maps from DexiNed (Soria et al., 2020). We evaluate semantic segmentation performance for image understanding on ADE-20K and Cityscapes (Cordts et al., 2016) datasets. Performance is measured with mIOU and FID between ground truth and generated masks.

**Implementation Details.** Our models are trained using TPU-v3 pods on Google Cloud. Our largest model, LVM-Lite-3B, completes training in ~10 days on a v3-256 TPU pod. In comparison, LVM-3B(Bai et al., 2023) requires 14 days to train on a more extensive v3-512 TPU pod.

**Video Generation.** As presented in Table 1, LVM-Lite demonstrates impressive video generation capabilities and remarkable scalability from 300M to 3B parameters, with the FVD score decreasing from **252** to **188** on UCF-101. We achieve comparable performance to CogVideo (Hong et al., 2023) with a much smaller model size. Compared to MagVit(Yu et al., 2023), LVM-Lite attains superior training efficiency with only 900 ImageNet-1K epochs, whereas MagVit requires fine-tuning computation equivalent to ~4000 ImageNet-1K epochs. Notably, our model handles a broader range of tasks without additional task-specific fine-tuning.

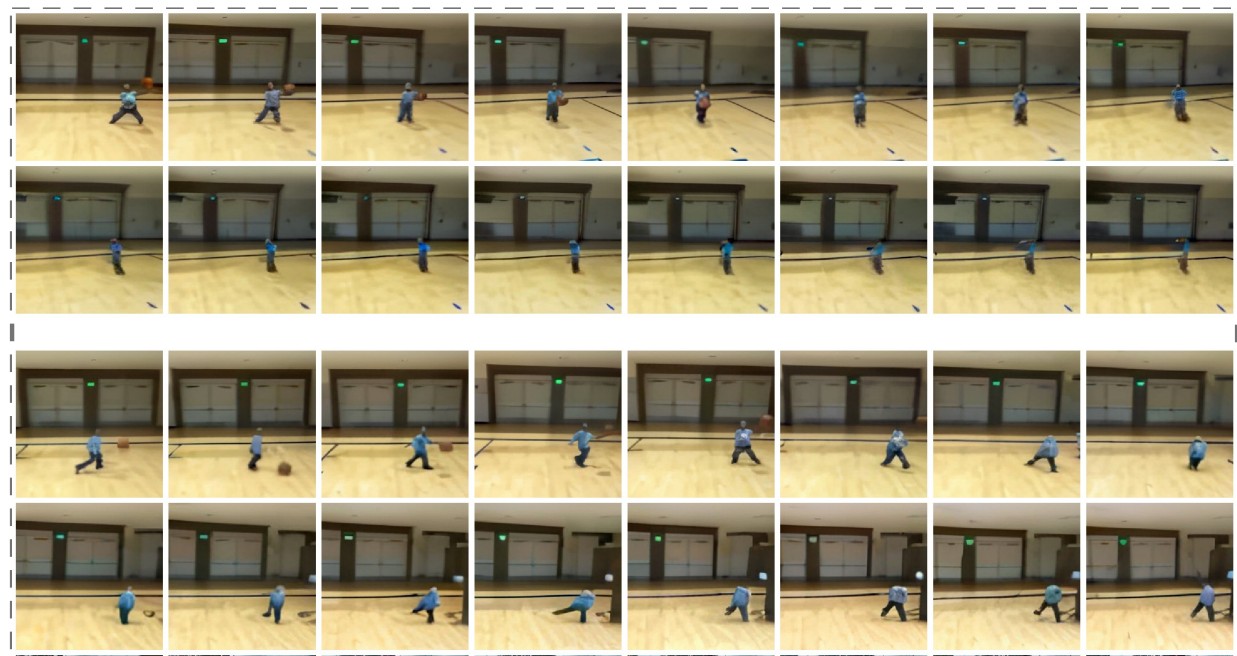

Figure 2: **Video generation on K600.** We generate high-fidelity videos with resolutions of $16 \times 256 \times 256$ using LVM-Lite-300M (top two rows) and LVM-Lite-3B (bottom two rows).

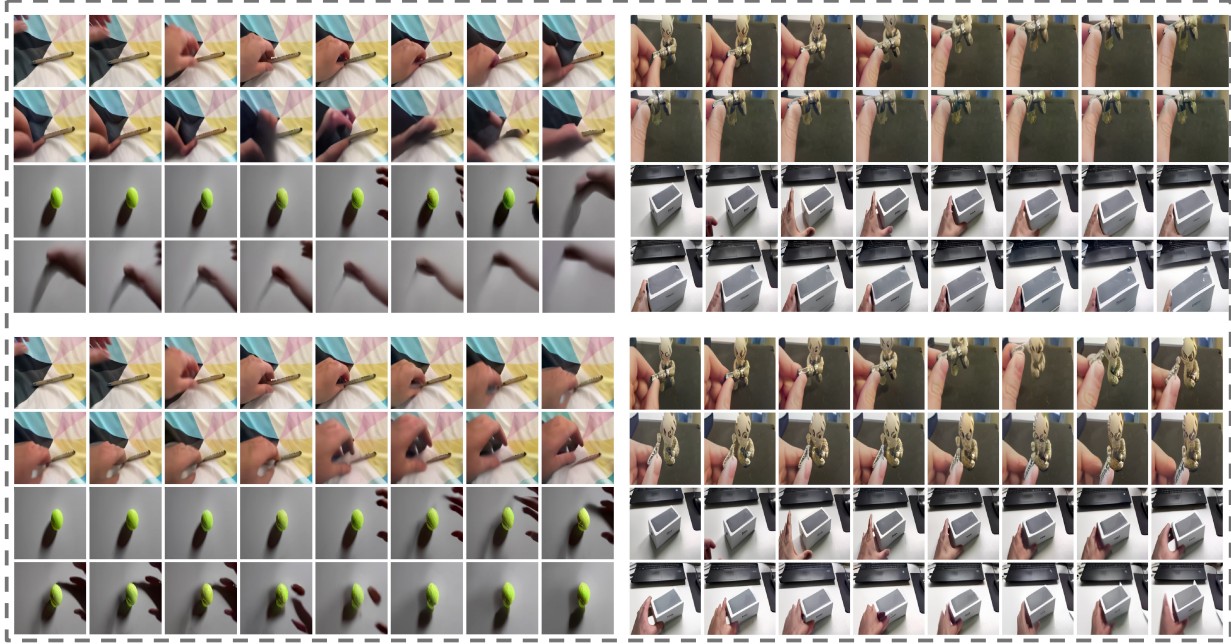

Figure 3: **Video generation on SS-V2.** LVM-Lite generates videos with high temporal consistency.

As depicted in the qualitative results from Figures 2 and 3, LVM-Lite produces higher resolution frames than previous efforts (Skorokhodov et al., 2021; Yu et al., 2023) and generates actions with high fidelity for motion-centric data in SS-V2.

**Image Generation and Understanding.** As presented in Table 2 and Figure 4, LVM-Lite achieves comparable performance with specialist models on image generation while using only a single training cycle. On ImageNet-1K, LVM-Lite generates realistic images from segmentation masks, edge maps, and depth

Table 3: **Comparison on image understanding.** We evaluate the image semantic segmentation task.

| Model | ADE20K-D | | City.-D |
|---|---|---|---|
| | mIOU ↑ | FID ↓ | mIOU ↑ |
| Mask2Former(Cheng et al., 2022) | 57.7 | - | 83.3 |
| Painter(Wang et al., 2023a) | 49.9 | - | N/A |
| LVM-lite-300M | 2.3 | 266.7 | 9.0 |
| LVM-lite-1B | 0.9 | 233.9 | 8.9 |
| LVM-lite-3B | 0.7 | 92.8 | 10.3 |
| Ground truth rec. | 35.2 | 93.8 | 58.6 |

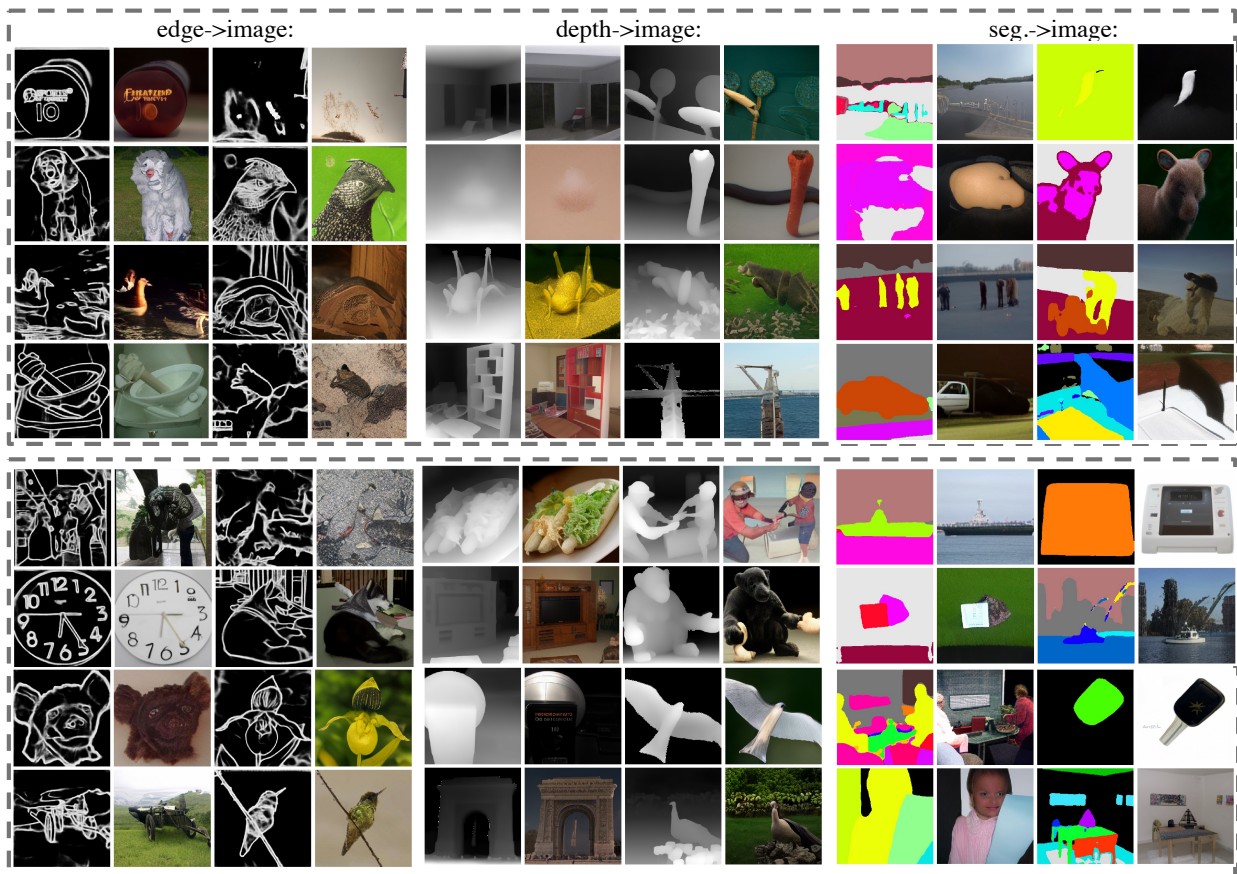

Figure 4: **Image generation on ImageNet-1K.** Top: 300M model. Bottom: 3B model. Our models can produce high-quality images across various modalities without fine-tuning on specific datasets.

maps. Model scaling also greatly enhances image generation, especially on ImageNet-1K, with LVM-Lite-3B outperforming LVM-Lite-300M by **23.0-28.6** in FID. Nonetheless, LVM-Lite faces challenges with image understanding as shown in Table 3, yielding lower mIOU compared with segmentation specialists. This modest segmentation performance might be partly attributed to the noise introduced in the tokenization of segmentation labels—performance is substantially lower even with reconstruction from ground truth tokens. This is compounded by the lack of pixel-to-pixel supervision, which is widely used in supervised specialists but not in next-token predictions in LVM.

Table 4: **Pre-training ablation.** We compare pretraining on single images vs random image sequences on three benchmarks and report the model forward TFLOPs under the batch size of 1. *We compare the total training wall clock time in core hours measured on 256 TPU-V3 with the same number of samples. Random image seq.: 16-random-image sequence pretraining. Singe image: single-image pretraining. We also report VQ-GAN's reconstruction performance using the ground truth as input.

| model size | random image seq. | single image | TFLOPs | speedup* | UCF-101 | | ADE20K-G | ADE20K-D | |
|---|---|---|---|---|---|---|---|---|---|
| - | | | | | FVD-16f ↓ | IS ↑ | FID↓ | mIOU↑ | FID↓ |
| 300M | ✔ | ✗ | 4.1 | ×1 | 274.6 | 33.8 | 59.3 | 1.7 | 103.5 |
| | ✗ | ✔ | 0.2 | ×**2.7** | 291.2 | 34.2 | 50.0 | 1.1 | 129.0 |
| 600M | ✔ | ✗ | 7.9 | ×1 | 268.7 | 34.6 | 44.8 | 1.0 | 129.4 |
| | ✗ | ✔ | 0.4 | ×**1.7** | 274.2 | 34.0 | 48.9 | 1.9 | 110.8 |
| 1B | ✔ | ✗ | 12.8 | ×1 | 256.0 | 34.5 | 44.5 | 0.9 | 136.6 |
| | ✗ | ✔ | 0.6 | ×**1.8** | 262.8 | 34.1 | 44.8 | 1.7 | 108.0 |
| 3B | ✔ | ✗ | 33.0 | ×1 | 212.3 | 34.2 | 45.3 | 1.1 | 125.9 |
| | ✗ | ✔ | 1.7 | ×**2.1** | 232.4 | 38.8 | 40.8 | 0.9 | 94.0 |
| ground truth rec. | | - | | | 119.0 | 35.9 | 20.4 | 35.2 | 93.8 |

Table 5: **Fine-tuning ablation**. We compare fine-tuning performance across different dataset combinations on a 300M model using the same compute budget.

| model size | fine-tuning datasets | | | | UCF-101 | | ADE20K-G | ADE20K-D | |
|---|---|---|---|---|---|---|---|---|---|
| - | natural seq. | generative seq. | discriminate seq. | random seq. | FVD-16f ↓ | IS ↑ | FID↓ | mIOU↑ | FID↓ |
| 300M | ✔ | ✗ | ✗ | ✗ | 284.0 | 34.3 | 187.0 | 0.3 | 132.3 |
| | ✗ | ✔ | ✗ | ✗ | 440.5 | 27.7 | 44.7 | 0.3 | 102.0 |
| | ✗ | ✗ | ✔ | ✗ | 842.9 | 15.2 | 120.0 | 1.3 | 116.0 |
| | ✗ | ✗ | ✗ | ✔ | 812.5 | 8.8 | 147.2 | 0.1 | 126.3 |
| | ✔ | ✔ | ✗ | ✗ | 299.2 | 34.2 | 46.1 | 0.5 | 117.7 |
| | ✔ | ✔ | ✔ | ✗ | 291.2 | 34.2 | 50.0 | 1.1 | 129.0 |
| | ✔ | ✔ | ✔ | ✔ | 372.6 | 37.8 | 75.7 | 0.5 | 91.8 |
| ground truth rec. | | - | | | 119.0 | 35.9 | 20.4 | 35.2 | 93.8 |

## 3.2 Ablation Study

We extensively ablate LVM-Lite to showcase its training efficiency and flexible task adaptation. We adopt the standard settings in Section 3.1 but reduce training budgets to 200-epoch pre-training and 100-epoch fine-tuning. We follow the same procedure outlined in Section 3.1 to evaluate video generation on UCF-101 (Soomro et al., 2012) and image generation and understanding on ADE-20K (Zhou et al., 2017).

**Single-image Pre-training.** Decoupling single-image pre-training from sequence fine-tuning is the core of LVM-Lite. We first ablate the effect of pre-training on single images vs random image sequences. As reported in Table 4, we observe a significant training acceleration, up to ×**2.7**, by pre-training on single images with reduced context length (from 4096 to 256) without any loss in performance. Notably, TFLOPs can be significantly reduced up to ×**20** with the same number of training tokens. Our proposed training method achieves performance comparable to pre-training with random image sequences for the three tasks. Thus, single-image pre-training is a good and efficient initializer for downstream tasks. However, a slight decrease in performance was noted in semantic segmentation, likely due to the lack of pixel-to-pixel supervision in next token prediction.

**Sequence Fine-tuning.** Next we ablate sequence fine-tuning, exploring the effects of different fine-tuning datasets, including natural sequences, discriminative sequences and generative sequences on various tasks. As presented in Table 5, and Figure 5, models fine-tuned on a specific data category excel in tasks related to that category. For example, models fine-tuned on natural sequences perform best in video generation with subpar performance in image generation and understanding. Default fine-tuning across all three categories

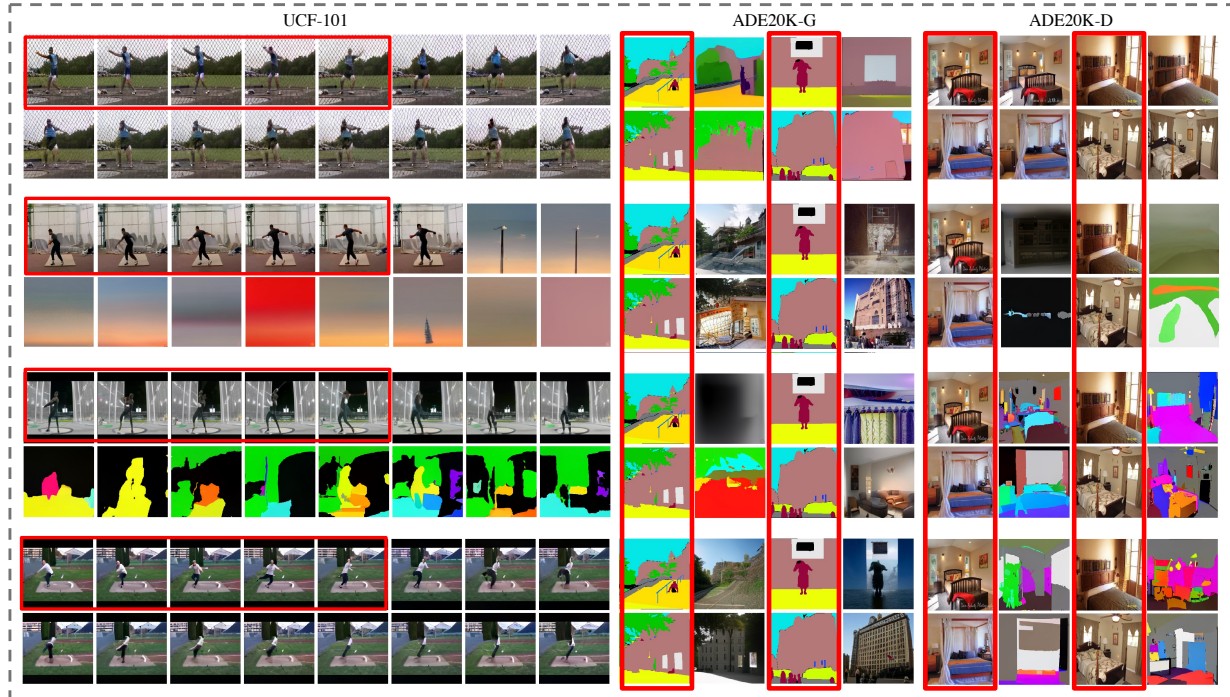

Figure 5: **Qualitative results on fine-tuning ablation.** Top two rows: fine-tuning on **natural sequences**. Mid two rows: fine-tuning on **generative sequences**. Next two rows: fine-tuning on **discriminative sequences**. Bottom two rows: fine-tuning on **all categories**. Prompts are not shown for image generation and segmentation. Red rec.: query frames, segmentation maps, or images.

Table 6: **Model & schedule scaling**. We adopt the strategy of equating seen images to the number of training epochs with ImageNet-1K(Deng et al., 2009) images to set controllable training budgets. Subsequently, with the training budget fixed, we scale the model size across three scales.

| model size | pre-training | fine-tuning | UCF-101 | | ADE20K-G | ADE20K-D | |
|---|---|---|---|---|---|---|---|
| | | | FVD-16f ↓ | IS ↑ | FID ↓ | mIOU ↑ | FID ↓ |
| | **200** | 100 | 291.2 | 34.2 | 50.0 | 1.1 | 129.0 |
| | **400** | 100 | 290.4 | 33.7 | 53.5 | 1.7 | 105.1 |
| 300M | 800 | **100** | 275.9 | 33.3 | 49.6 | 1.8 | 107.3 |
| | 800 | **200** | 263.8 | 34.4 | 44.2 | 1.4 | 119.1 |
| | 800 | **300** | 252.2 | 33.7 | 41.2 | 2.3 | 266.7 |
| **300M** | 800 | 300 | 252.2 | 33.7 | 41.2 | 2.3 | 266.7 |
| **1B** | 800 | 300 | 222.7 | 34.9 | 42.4 | 0.9 | 233.9 |
| **3B** | 800 | 300 | 188.2 | 33.8 | 39.7 | 0.7 | 92.8 |
| ground truth rec. | - | - | 119.0 | 35.9 | 20.4 | 35.2 | 93.8 |

emerges as a versatile solution capable of effectively addressing both generative and discriminative tasks. In particular, we note that the inclusion of random image sequences during fine-tuning significantly degrades performance across all tasks, reinforcing that random image sequences introduce noise with a detrimental effect on training.

**Scalability.**  First, we establish the schedule scalability of LVM-Lite. As illustrated in Table 6, increasing pre-training epochs enhances downstream task performance. 800-epoch pre-training significantly augments video generation performance compared to 200-epoch pre-training **(291.2 vs. 275.9)**. Extending fine-tuning schedule from 100 to 300 epochs also improves performance further by **23.7**. Next, we demonstrate the model scalability of LVM-Lite. As presented in Table 6 and Figures 2 and 4, a 3B model markedly outperforms a

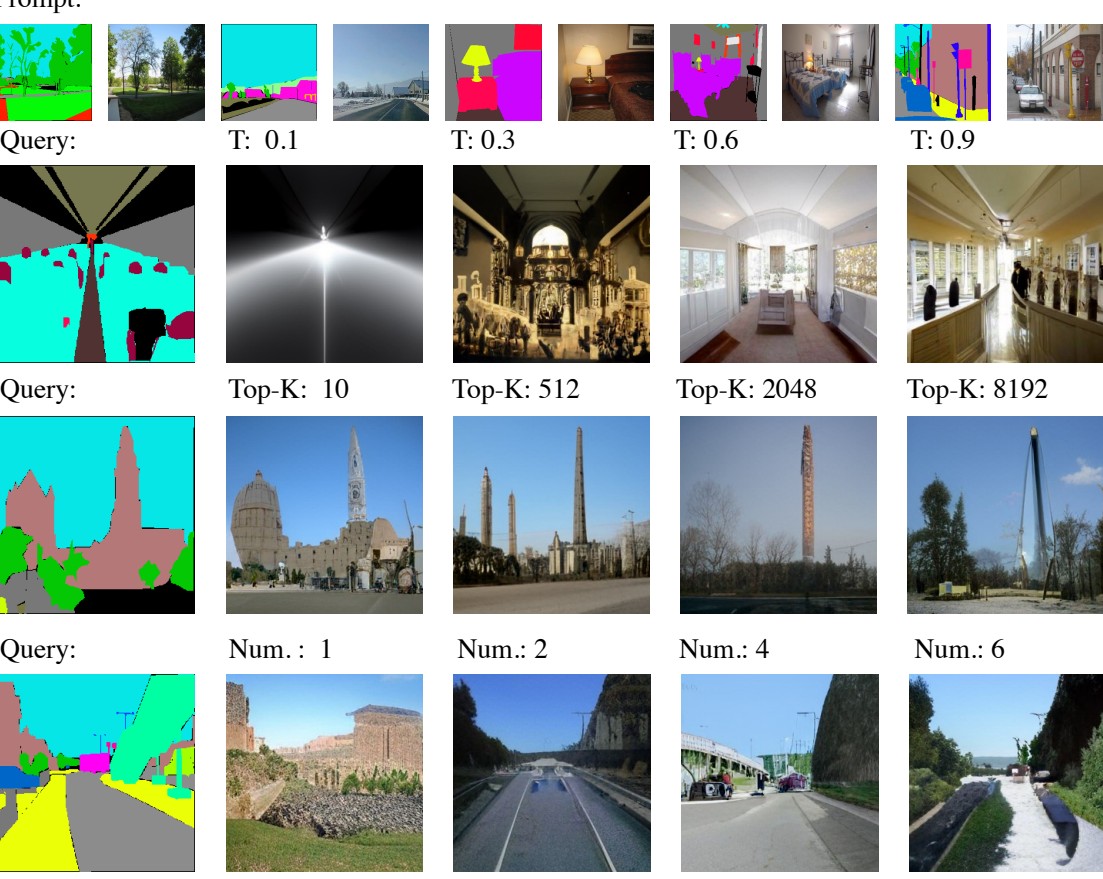

Figure 6: **Evaluation hyper-parameters.** The influence of temperature (T, from 0.1 to 0.9), Top-K (from 10 to 8192), and the number of prompts (Num. from 1 to 6) on the evaluation performance.

300M model on video and image generation. For instance, LVM-Lite-3B surpasses LVM-Lite-300M by **64** on the FVD score on UCF-101. Scaling benefits video generation more than image understanding, likely due to the decoder-only architecture being more effective and the VQ-GAN appearing better suited for generation tasks.

**Evaluation parameters** We examine the impact of varying three key parameters for evaluation: the number of prompts from 1 to 6, the top-k range from 100 to 8192, and the temperature from 0.1 to 0.9. We focus on the conditional image generation task on ADE20K. As shown in Figure 6, these parameters are crucial for modulating the diversity and creativity of the generated outputs. Aligned with observations in LLMs, a lower temperature and top-k setting produces more coherent results, while a higher temperature and top-k encourage greater creativity and novelty in the generative process. We also find that generated images are more realistic as the number of prompts increases.

## 4 Related Works

**Pre-trained Vision Models.** Before the advent of LLMs, pre-trained vision models serve as improved feature extractors for downstream tasks like image classification (Sun et al., 2017), object detection(Girshick et al., 2015; Girshick, 2015; Ren et al., 2015), video classification(Bertasius et al., 2021; Arnab et al., 2021; Li et al., 2022; Fan et al., 2021) and semantic segmentation(Cheng et al., 2022; Ren et al., 2015; Long et al., 2015; He et al., 2017). Self-supervised learning (Doersch et al., 2015; Chen & He, 2020; Chen et al., 2020a; He et al., 2019; Chen et al., 2020b) share the same goal and recent proposed masked auto-encoder (He et al., 2021; Tong et al., 2022; Bao et al., 2021) based methods showcase outstanding scaling ability of vision

transformers. Natural language supervision (Radford et al., 2021; Jia et al., 2021; Li et al., 2023b) is an effective scaling-up method for various vision downstream tasks. Contrary to approaches that offer merely a vision backbone, our work develops a vision generalist akin to LLMs, designed to undertake downstream tasks without extensive fine-tuning.

**Visual in-context Learning.** Language models, exemplified by GPT-3 (Brown et al., 2020), have excelled in in-context learning. Similarly, vision-language models (Liu et al., 2023; Alayrac et al., 2022; Jaegle et al., 2021; Li et al., 2023a) demonstrate this capability leveraging vast datasets. Close to another trend of visual Prompting techniques (Bar et al., 2022; Wang et al., 2023b; Wu et al., 2023; Wang et al., 2023a;c), our work further enhances the performance by utilizing visual cues to guide learning and task interpretation. We emphasize the potential of large-scale training to enable in-context learning ability.

## 5 Conclusion

Generative pre-training has inspired innovations in visual understanding through large-scale pre-training on visual sequences. Our work introduces a novel two-phase decoupled learning approach, enhancing training efficiency without sacrificing performance. Our proposed LVM-Lite excels in video prediction and conditional image/video generation, demonstrating comparable performance to specialist models across benchmarks. By evaluating LVMs' in-context learning capabilities comprehensively on various benchmarks, we show that our LVM-Lite can successfully speed up the whole training process and demonstrate strong scalability.

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

# A  Appendix / supplemental material

## A.1  Implementation details

**Model configuration.**  In our experiments, we systematically explore four models whose configurations are listed in Table 7. These models are based on a decoder-only architecture, specifically leveraging the Llama-2 framework (Touvron et al., 2023), chosen for its efficiency and adaptability to our framework. Due to limited computation resources, our largest 3B model adopted an advanced block-parallel transformer(Liu & Abbeel, 2023) to reduce memory requirements further. All of our experiments are conducted on a 256-core TPU-v3. Our implementation is based on JAX(Bradbury et al., 2018)

Table 7: Model architecture

| model size | hidden dim | MLP dim | heads | layers |
|---|---|---|---|---|
| 300M | 1024 | 2688 | 8 | 22 |
| 600M | 1536 | 4096 | 16 | 22 |
| 1B | 2048 | 5504 | 16 | 22 |
| 3B | 3200 | 8640 | 32 | 26 |

Table 8: Hyperparameters for pre-training and fine-tuning.

| hyperparameter | single-image pre-training | sequence fine-tuning |
|---|---|---|
| learning rate schedule | linear warmup and cosine decay | |
| weight decay | 0.1 | |
| optimizer | AdamW(Loshchilov & Hutter, 2019) | |
| optimizer momentum | $\beta_1 = 0.9, \beta_2 = 0.95$ | |
| base learning rate | 1.5e-4 | 1.5e-5 |
| final learning rate | 1.5e-5 | 1.5e-6 |
| warmup steps | 2000 | 0 |
| total training steps | 125112 | 15639 |
| batch size | 8192 | 512 |
| context length | 256 | 4096 |

## A.2  Training and evaluation.

We also provide detailed pre-training and fine-tuning hyperparameters in Table 8. We use training hyperparameters based on (Bai et al., 2023). To enhance efficiency, we ensure that the total number of processed tokens per iteration remains constant, increasing the pre-training batch size by ×16. For our evaluation, we utilize prompts to specify tasks in line with (Bai et al., 2023). Instead of employing seven pairs of images, we discovered that a single pair is adequate for task indication. These prompts are illustrated in Figure 7. Our approach allows us to assess qualitative and quantitative results across different tasks. We set the Top-K as 100 and the temperature as 1. For the video prediction task. we sample 16 frames from the original video to be the ground truth and use the first 5 frames as prompt. We ask the model to generate the rest 11 frames. For ImageNet-1K(Deng et al., 2009) generation, we evaluate our model on the validation set. We randomly sample one of the training set prompts and use the same prompt for all validation images.

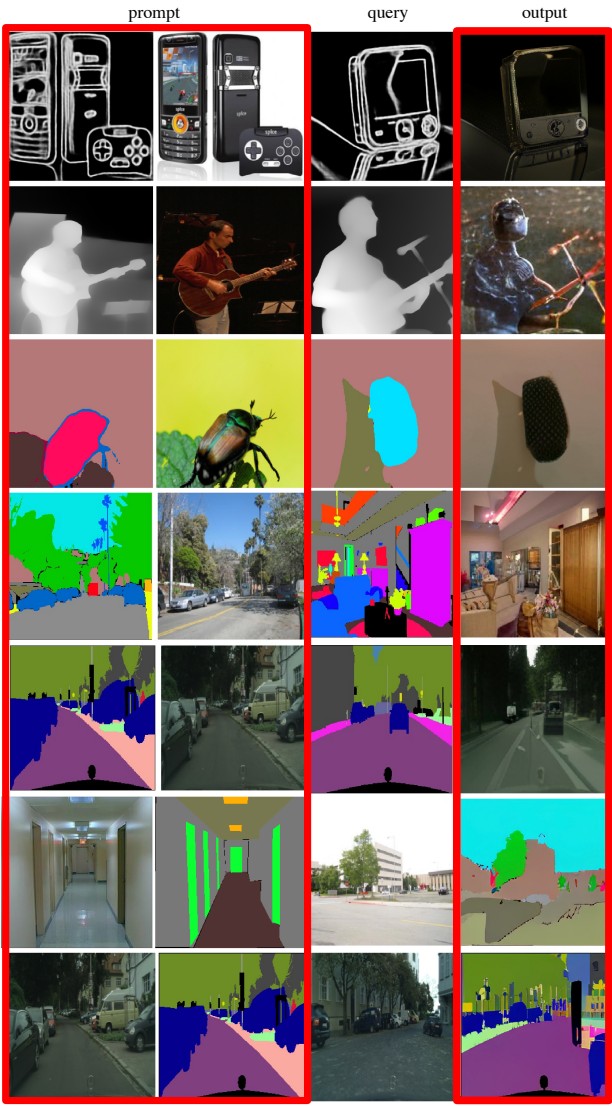

Figure 7: Example of prompts in our metric-evaluation. We use a single prompt containing one image pair to indicate a task.

We use (Soria et al., 2020) to infer the edges map, mask2former(Cheng et al., 2022) to generate semantic masks, and depth-anything(Yang et al., 2024) to predict depth map. For ADE20K(Zhou et al., 2017) and Cityscapes(Cordts et al., 2016) generation task, we use ground truth segmentation map to be the condition. For each task, we first adjust the image size to $256 \times 256$ using bilinear interpolation and apply the nearest neighbor interpolation method to resize the masks.

## A.3 Datasets

Here, we present in detail how we construct our datasets. For training, we mainly follow LVM(Bai et al., 2023) to construct our datasets; we pre-process most datasets listed in (Bai et al., 2023) in the same manner. We list all datasets used in our experiments in Table 9. We always keep 16 images for different tasks as the length of image sequences. Thus, for video generation task, we sample 16 frames from the original video. For image-based tasks, we use 8 image pairs to form a single image sequence. For evaluation, we show the details of datasets we used in Table 10.

Table 9: Full training dataset. We follow LVM(Bai et al., 2023) to construct datasets but divide them into generative and discriminative tasks.

| dataset | task type | annotation source |
|---|---|---|
| **random image sequence** | | |
| DataComp-1B(Gadre et al., 2023) | inpainting | ground truth |
| **natural sequence** | | |
| UCF101 (Soomro et al., 2012) HMDB (Kuehne et al., 2011) Moments in Time (Monfort et al., 2019) Multi-moments in Time (Monfort et al., 2021) Co3D (Reizenstein et al., 2021) Charades v1 (Sigurdsson et al., 2016) Something-something v2 (Goyal et al., 2017a) Kinetics 700 (Carreira et al., 2019) Jester (Materzynska et al., 2019) MultiSports (Li et al., 2021) CharadesEgo (Sigurdsson et al., 2018) AVA (Murray et al., 2012) Ego4D (Grauman et al., 2022) Objaverse (Deitke et al., 2023) Rendered Multiviews | video generation | ground truth |
| **generative sequence** | | |
| ImageNet-1K (Deng et al., 2009)



COCO (Lin et al., 2014) ADE 20K (Zhou et al., 2019), Cityscapes (Cordts et al., 2016) Subset of InstructPix2Pix (Geiger et al., 2012) Charades V1 (Sigurdsson et al., 2018) VIPSeg (Miao et al., 2022) Co3D (Reizenstein et al., 2021) Co3D (Reizenstein et al., 2021) | image to image segmentation map(Cheng et al., 2022) to image depth map (Yang et al., 2024) to image edge map (Soria et al., 2020) to image inpainting colorization instance segmentation to image segmentation map to image style transfer segmentation map(Cheng et al., 2022) to video panoptic segmentation to video object mask to video depth to video | ground truth |
| **discriminative sequence** | | |
| COCO (Lin et al., 2014) | object detection | - |
| ADE20K (Zhou et al., 2019), Cityscapes (Cordts et al., 2016) | semantic segmentation | ground truth |
| ImageNet-1K (Deng et al., 2009) | semantic segmentation | Mask2Former (Cheng et al., 2022) |
| COCO (Lin et al., 2014) | human pose | ground truth |
| COCO (Lin et al., 2014), ImageNet-1K (Deng et al., 2009) | depth map image | Depth-anything (Yang et al., 2024) |
| COCO (Lin et al., 2014), ImageNet-1K (Deng et al., 2009) | edge detection | DexiNed (Soria et al., 2020) |
| SIDD (Abdelhamed et al., 2018) | denoised image | ground truth |
| LOL(Wei et al., 2018) | light-enhanced image | ground truth |
| VIPSeg (Miao et al., 2022) | video panoptic segmentation | ground truth |
| VOS (Xu et al., 2018b) | video object segmentation | ground truth |
| Co3D (Reizenstein et al., 2021) | video object segmentation | ground truth |
| Co3D (Reizenstein et al., 2021) | video object segmentation | ground truth |

Table 10: Evaluation datasets and metrics used for comparison for Table 4.

| dataset | split&number of samples | metric |
|---|---|---|
| **frame prediction** | | |
| UCF101 (Soomro et al., 2012) | test & 3783 | FVD&IS |
| Something-something v2 (Goyal et al., 2017a) | validation & 24777 | FVD |
| Kinetics 600 (Carreira et al., 2019) | validation & 31593 | FVD |
| **image synthesis** | | |
| ImageNet-1K (Deng et al., 2009) | validation & 50000 | FID |
| ADE20K (Zhou et al., 2019) | validation & 2000 | FID |
| Cityscapes (Cordts et al., 2016) | validation & 500 | FID |
| **semantic segmentation** | | |
| ADE20K (Zhou et al., 2019) | validation & 2000 | mIOU&FID |
| Cityscapes (Cordts et al., 2016) | validation & 500 | mIOU&FID |

### A.4 Additional Results

Our additional quality evaluation spans tasks on COCO(Lin et al., 2014), ImageNet-1K(Deng et al., 2009), and VIPSeg(Miao et al., 2022) datasets shown in Figure 8,9,10 and 11. We demonstrate our model's capability on COCO in human pose estimation and object detection. We utilize (Brooks et al., 2023) for style transfer to showcase our approach's adaptability. On ImageNet-1K, we cover edge detection, inpainting, semantic segmentation, relative depth estimation, and colorization, illustrating the model's versatility across different image processing challenges. Additionally, VIPSeg's qualitative results are included, where the task involves generating frames from 8 object masks, highlighting our model's proficiency in image synthesis.

**Long-video Generation.** We enhance LVM-Lite's capability to generate longer videos by fine-tuning it with the SS-V2 dataset to process 64 frames after 1500 iterations. We highlight the proficiency of LVM-Lite to generate high-quality, extended sequences sequences in Figure 12.

### A.5 Limitation and Broader Impacts

As discussed in Section 3.1, while our model shows excellent scalability, high-quality generation capabilities, and general task awareness, its performance on discriminative tasks, such as semantic segmentation, remains significantly lower compared to current state-of-the-art in-domain models. This modest segmentation performance may be partly due to the noise introduced during the tokenization of segmentation labels—performance remains substantially lower even with reconstruction from ground truth tokens. Additionally, the lack of pixel-to-pixel supervision, commonly used in supervised specialist models, further compounds the issue, as it is not employed in next-token prediction within LVM. Addressing this issue is beyond our current scope, as our focus is on efficiency and providing a comprehensive study on training effective LVM. We plan to leave this as future work.

Since this paper focuses on democratizing the training burden of current large vision models, we believe that migrating the training difficulty can help researchers reduce their research cycles and dedicate more efforts to developing robust novel methods. However, this paper's potential negative social impact is that our generative model might produce content using harmful or privacy-concerning training data that may be overlooked. To mitigate this, we will rigorously test our model and consider implementing gated access for safety concerns.

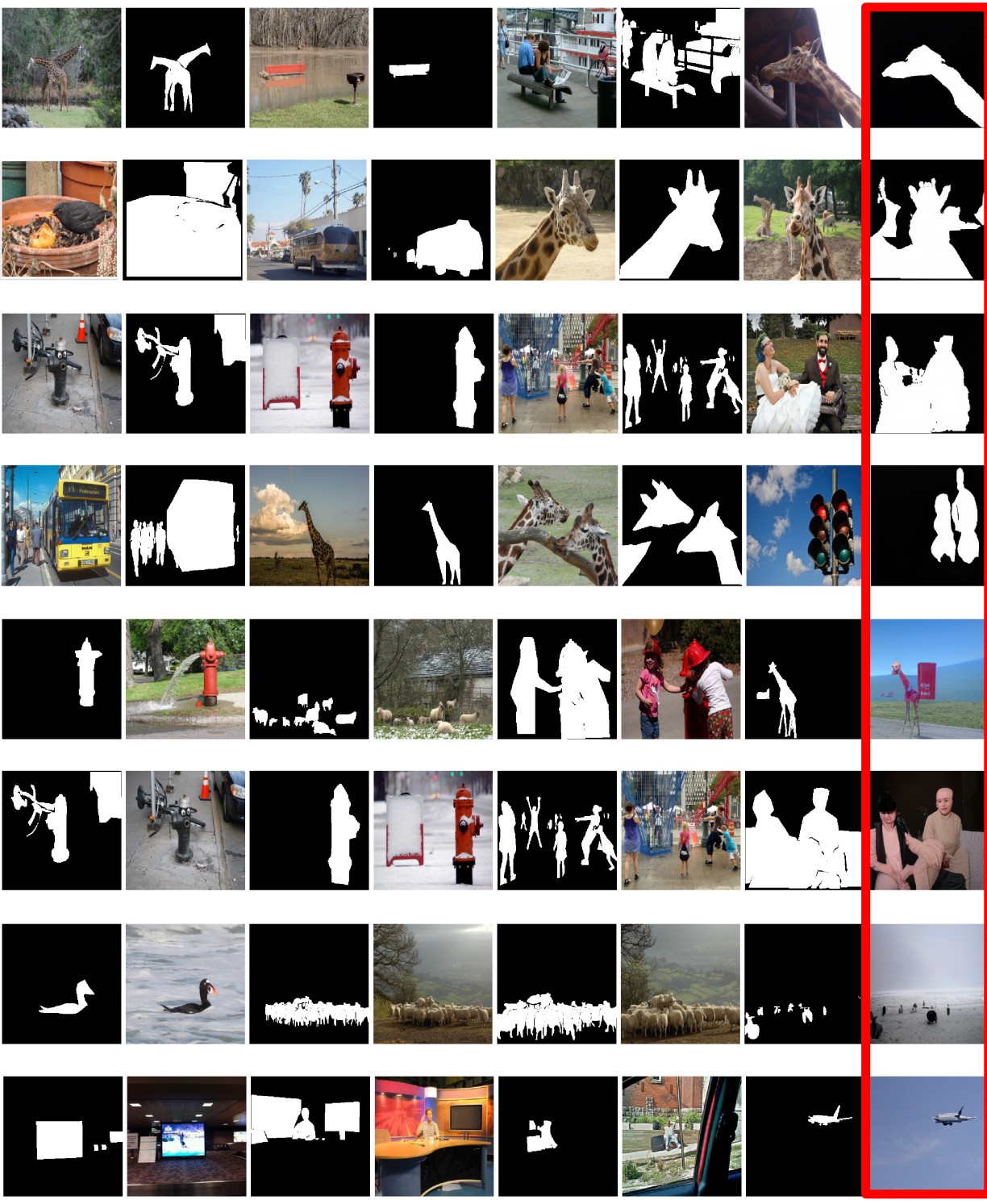

Figure 8: COCO(Lin et al., 2014) evaluation. Red: the generated results. First four rows: segmentation task. Second four rows: segmentation to images. For all examples, we use three prompts that contain 6 images in total to indicate the task and one query mask/image.

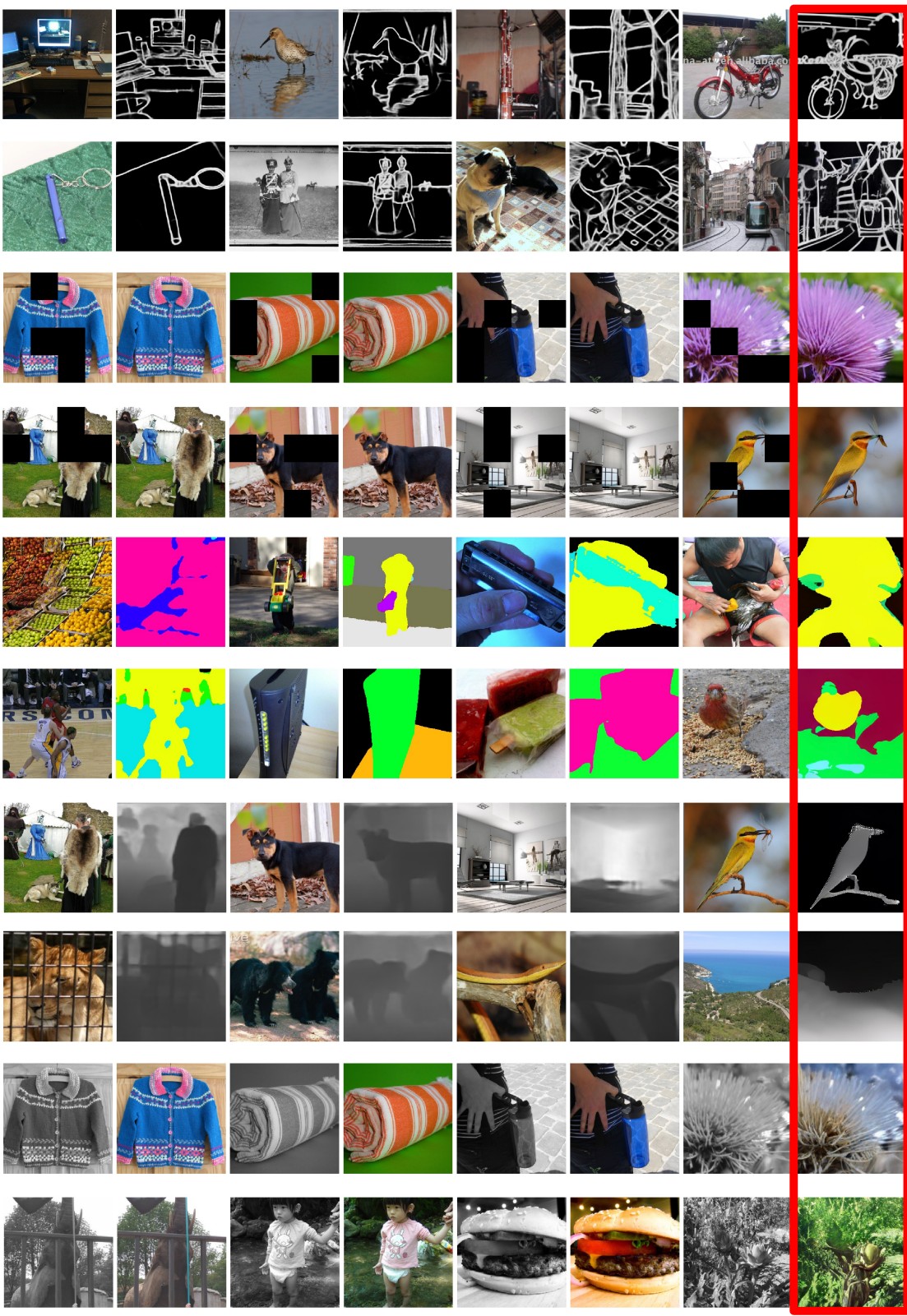

Figure 9: ImageNet-1K(Deng et al., 2009) qualitative evaluation on validation set. Red: the generated results. For all examples, we use three prompts that contain 6 images in total to indicate the task and one query. We show edge detection, inpainting, semantic segmentation, relative depth estimation, and colorization tasks.

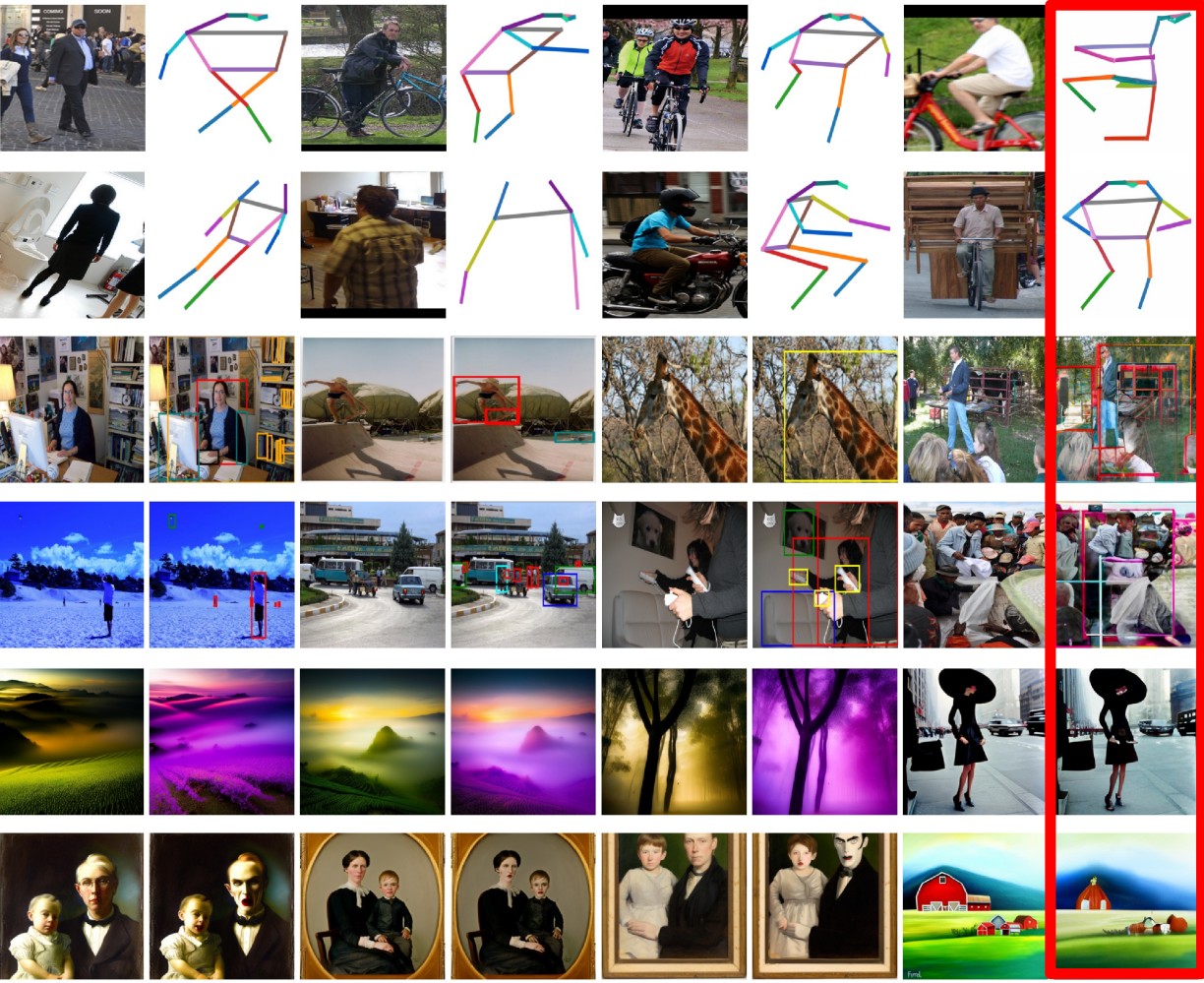

Figure 10: Other tasks' qualitative evaluation set. Red: the generated results. For all examples, we use three prompts that contain 6 images in total to indicate the task and one query. We show human pose estimation, object detection and style transfer tasks.

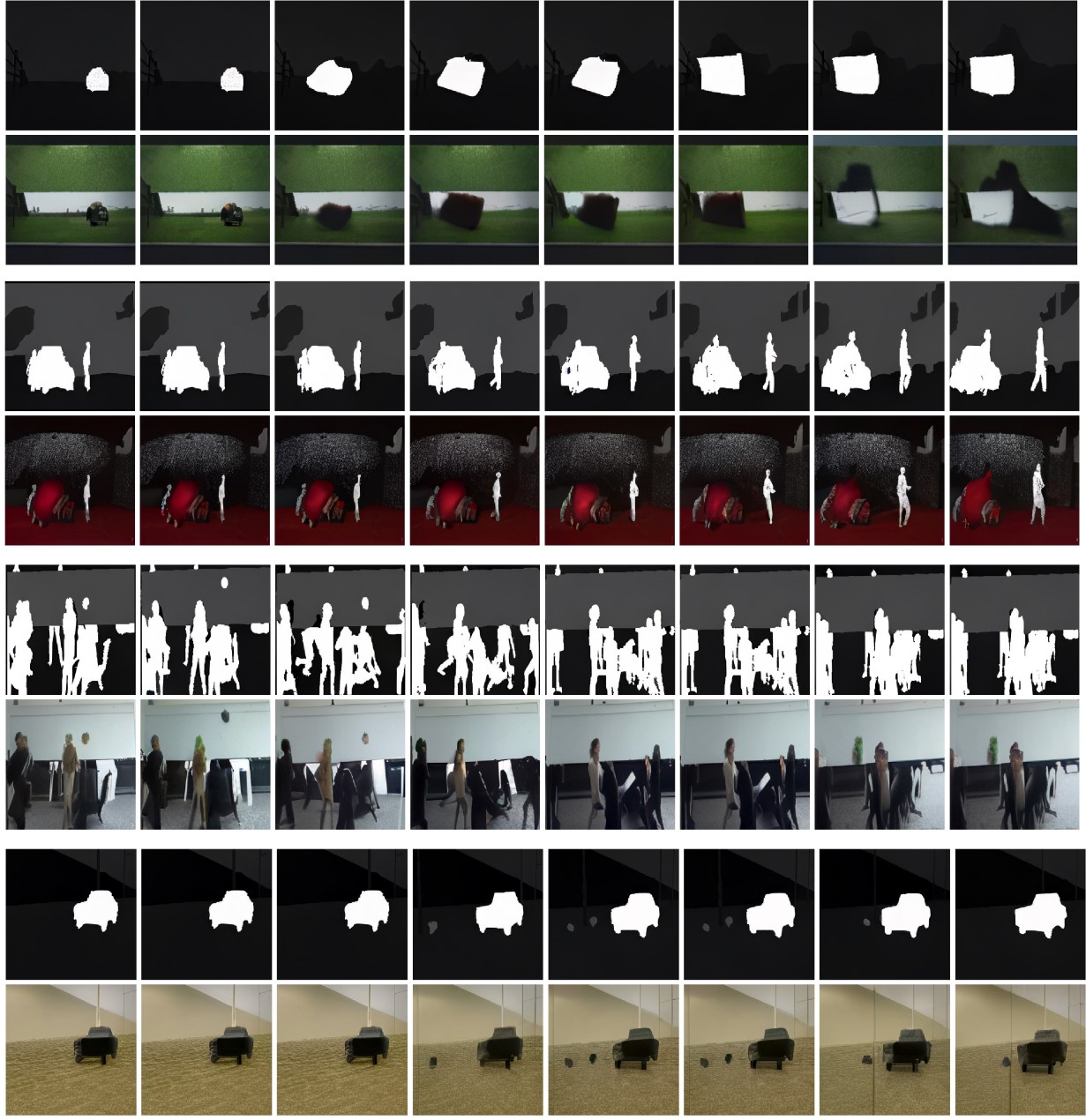

Figure 11: VIPSeg(Miao et al., 2022) qualitative results. Task: given 8 masks, generate the corresponding frames.

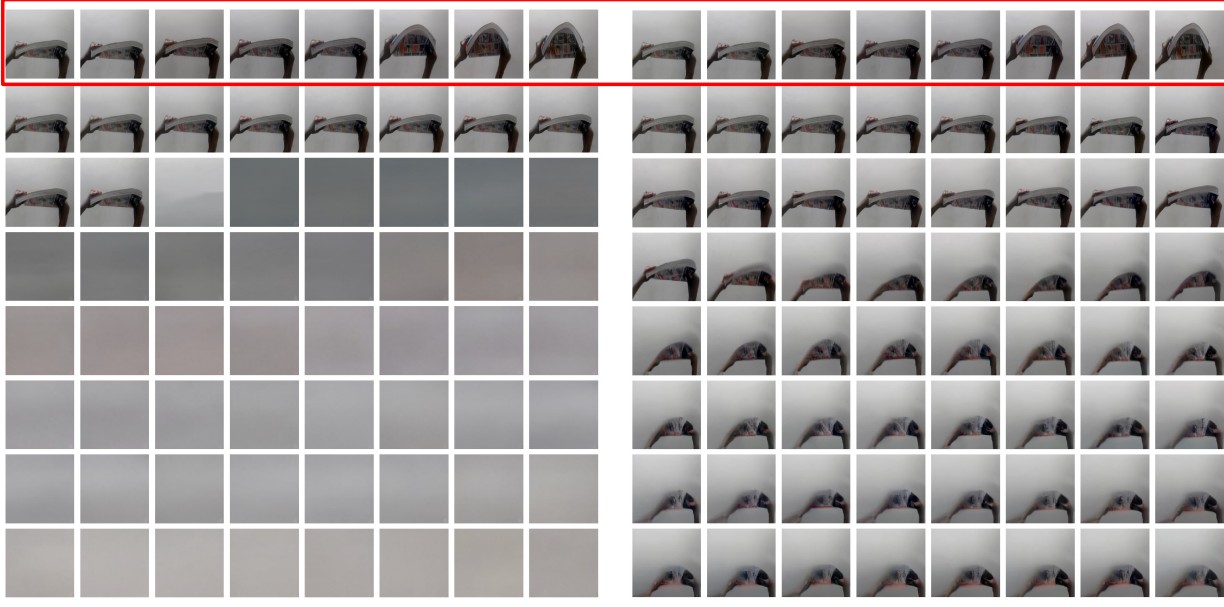

Figure 12: 64-frame video generation. Left: default train with 16-frame (4096 context length). Right: extended 64-frame (16K context length). Red rec.: a short action clip prompt. Task: predict the next 60 frames. Spatial resolution:$256 \times 256$.

