# OpenReview forum: "LVM-Lite: Training Large Vision Models with Efficient Sequential Modeling"
_TMLR — Rejected by TMLR_

### Review · Reviewer_HkoL · 2024-10-05

**Summary Of Contributions:**

The paper proposes an efficient training recipe for autoregressive vision models. The contribution consists of training on single images instead of sequences of random images, as done in the stage-1 training of LVM. The work shows competitive performance with significant reduction in training cost.

**Audience:**

Yes

**Broader Impact Concerns:**

The authors discuss the broader impact in the appendix.

**Claims And Evidence:**

No

**Requested Changes:**

In addition to the points raised in the weaknesses sections. I have the following questions that might need further clarifications:

* How do the authors change the hyperparameters (e.g. learning rate) when scaling the model size.
* Is the stage-1 training is important for the final model? Do the authors tried to train the model on the downstream datasets for example for longer iterations and compare the results of their current model?
* Did the authors notice any form of task transfer when doing multitask fine tuning?

**Strengths And Weaknesses:**

**Strengths:**
* The paper is well written and clearly presented
* The method leads to significant reduction in training cost
* The work is evaluated on wide range of downstream tasks

**Weaknesses:**
* Novelty: while this should not be cause for rejection in itself, it will hinder the impact of this work. The contribution is simply replacing training on sequences of multiple images to training on single ones. Multi-stage training on single images followed by sequence of images is widely adopted as in [1, 2], and the authors don't discuss such related works.
* The paper proposes a more efficient training approach compared to LVM, however, there is no comparison to this baseline in the paper on downstream task performance.
* One main element in the proposed model is ICL. However, there is no proper ICL quantitative evaluation. For example, how does the performance change with the number of ICL shots?
* The model is significantly worse than other baselines on image understanding tasks (Tab. 3). It seems that the model is only good for generation and the authors should refine their claims accordingly (e.g. last sentence in the introduction)

[1] Shukor, Mustafa, et al. "Unified model for image, video, audio and language tasks." TMLR (2023).

[2] Wang, Junke, et al. "Omnivl: One foundation model for image-language and video-language tasks." Advances in neural information processing systems 35 (2022): 5696-5710.

---

> ### Author Response · Authors · 2024-11-29
> **Response to Reviewer HkoL**
>
> We appreciate Reviewer HkoL’s feedback on our initial manuscript, and we hope we have addressed their questions adequately as detailed below.
>
>
>
> **Q1:** Novelty: while this should not be cause for rejection in itself, it will hinder the impact of this work. The contribution is simply replacing training on sequences of multiple images to training on single ones. Multi-stage training on single images followed by sequence of images is widely adopted as in [1, 2], and the authors don't discuss such related works.
>
> **A1:**
> We thank Reviewer HkoL for suggesting these important and relevant references. These works align with the motivation of our approach. We address a specific limitation in LVM, focusing on the extensive use of random image sequences with inherently higher noise. We will include more discussions on these relevant works in the introduction.
>
> ---
>
> **Q2:** The paper proposes a more efficient training approach compared to LVM, however, there is no comparison to this baseline in the paper on downstream task performance.
>
> **A2:**
> We thank Reviewer HkoL for their rigor in the comparison with LVM baseline. We include new comparisons using the officially released checkpoint from LVM in the newly submitted Supplementary Material. At the time of writing, the official checkpoint from LVM has not been released and thus we did not include it in our initial manuscript.
>
> ---
>
> **Q3:** One main element in the proposed model is ICL. However, there is no proper ICL quantitative evaluation. For example, how does the performance change with the number of ICL shots?
>
> **A3:**
> Thank you for this question, please refer to the newly submitted Supplementary Material for details.
>
> ---
>
> **Q4:** The model is significantly worse than other baselines on image understanding tasks (Tab. 3). It seems that the model is only good for generation and the authors should refine their claims accordingly (e.g. last sentence in the introduction).
>
> **A4:**
> We agree with Reviewer HkoL that LVM-Lite still has a substantial performance gap on image understanding tasks, and we will revise the manuscript to tone down the rhetoric.
>
> ---
>
> ### Requested Changes
>
> **Q5:** How do the authors change the hyperparameters (e.g. learning rate) when scaling the model size?
>
> **A5:**
> We apologize for the lack of important training details and we will include them in the revised manuscript.
>
> ---
>
> **Q6:** Is the stage-1 training important for the final model? Do the authors try to train the model on the downstream datasets, for example, for longer iterations and compare the results of their current model?
>
> **A6:**
> Thank you for the suggestion. Please refer to our response to Reviewer tkS4 Weakness 1 for more details.
>
> ---
>
> **Q7:** Did the authors notice any form of task transfer when doing multitask fine-tuning?
>
> **A7:**
> Thanks for raising this question. We noticed some level of task transfer, as shown in Table 5, where LVM-Lite fine-tuned on a subset of the sequences still generalizes to other tasks. However, we observe the best performance when all categories (natural, generative, and discriminative sequences) are included in the fine-tuning datasets, except for the random image sequences which show detrimental effects on task performance.

---

### Review · Reviewer_2Ntn · 2024-10-08

**Summary Of Contributions:**

This paper presents LVM-Lite, a visual generative pretraining method that builds upon LVM, a previously published method. LVM-Lite brings computational improvements compared to LVM, with a novel two-stage pretraining, first stage on single images instead of random sequences of images, second on video sequences. LVM-Lite is evaluated on a series of images and video prediction benchmarks with in-context evaluation, and demonstrates good quantitative and qualitative performance while being more computationally efficient than LVM.

**Audience:**

Yes

**Broader Impact Concerns:**

The paper has a broader impact statement describing the potential negative social impact of generative models.

**Claims And Evidence:**

No

**Requested Changes:**

Provide qualitative, and quantitative if possible, comparisons with LVM on the same prompts to show that the model is of similar quality and that the computational gains are not at the cost of the quality of the model. This is essential validating the approach, and for me to recommend the paper for acceptance.

**Strengths And Weaknesses:**

This paper proposes a direct incremental improvement over LVM [1] and should therefore be compared to it both for the positive and negative aspects.

Strengths:

- Compared to LVM, the authors took the time to evaluate the model quantitatively on video prediction benchmarks on UCF-101, SSv2, and K600, and on images on ImageNet, ADE20k and Cityscapes. These metrics are more concrete than the perplexity and allow to compare the model to other generative approaches. The model scales well with model size and gives reasonable results compared to other approaches on these benchmarks.

- The training speed is significantly improved, instead of training on random sequences of 16 images mixed with video sequences, so 256 * 16 tokens, they train first on single images so 256 tokens, then fine-tune on video sequences. Table 4 shows that the speedup on the total training size is between x1.7 and x2.7, depending on model size, for a similar performance on downstream benchmarks. This is expected as the random image sequence of LVM was clearly not optimal, but nice to see in practice.

- The presentation is good and the ideas the authors want to communicate are very clear.

Weaknesses:

- The main issue of the paper is that it does not provide an apple to apple comparison with LVM on both quantitative and qualitative results.

Quantitative: LVM does not have a lot of quantitative results. Would it be possible to evaluate LVM on the same benchmarks ? Or having the same evaluation as Table 1 in [1]. My understanding is that the comparison of Table 4 is a reproduction of the authors, but is the baseline they reproduced of the same quality ?

Qualitative: The overall quality of the examples shown is not really good, in Figure 2, 3 and 5 the quality rapidly degrades as the model generates additional frames, and in Figure 5 and 8 the quality of the generated masks is very low. It seems that the examples in LVM are of better quality. Without direct comparison with LVM on the same prompts it is hard to conclude if this model is as good. Also LVM evaluates on a much wider diversity of prompts, such as sketches, surface normal estimation or grayscale to color.

- The performance on video prediction benchmarks is not very impressive, and not at the level of SOTA for both video generation compared to MAGVIT and images compared to VQ-GANS. What is the advantage of this class of method (autoregressive generative) compared to other more specialized models ?

- I don’t really see the point of measuring mIoU of generated segmentation maps when the performance is around ~1 or 2 points. I don’t think the comparisons are meaningful and maybe the qualitative examples are enough to judge the quality. The authors claim that the model has “impressive performance across various generative and discriminative” tasks, but does discriminative mean the results on segmentation of Table 3 ? Another interestting discriminative evaluation could be to probe the model on representation learning tasks, such as linear evaluation, linear segmentation of the learned features.

- The conclusion that this approach scales with size was already found in [1]. Here the main addition of the paper is to focus on computational efficiency so it would be interesting to study the sample-efficiency. Given various fixed computational budgets, what performances can you obtain compared to LVM ?

- The speedup is only for training time and not inference. Training time is usually less of an issue for practical applications. Have you thought about also improving the architecture to improve inference time ?

Questions:

- In Table 3, a better FID correlates to a worse mIoU, do you have an explanation for that ?
- Some models have a better FID or IS score compared to the “ground truth rec.” is it some form of overfitting ? How are these metrics reliable compared to just looking at the generated examples ?

[1] Sequential Modeling Enables Scalable Learning for Large Vision Models,  Bai et. al.

---

> ### Author Response · Authors · 2024-11-29
> **Response to Reviewer 2Ntn (Part 1)**
>
> We thank Reviewer 2Ntn for their nice summary of our study, and we hope we have addressed their questions adequately, as detailed below.
>
>
> **Q1:** The main issue of the paper is that it does not provide an apple-to-apple comparison with LVM [...] Also, LVM evaluates on a much wider diversity of prompts, such as sketches, surface normal estimation, or grayscale to color.
>
> **A1:**
> Thanks for the suggestions. We include new quantitative comparisons with LVM using the officially released checkpoints in the newly submitted Supplementary Material. Please refer to our response to Reviewer tkS4 Weaknesses #1-3 and the newly submitted Supplementary Material for details on the comparison results.
>
> We note that LVM-Lite is also evaluated on a diverse set of prompts, including edge maps, grayscales, etc. Please see Figures 8-12 for more examples.
>
>
> ---
>
> **Q2:** The performance on video prediction benchmarks is not very impressive. [...] What is the advantage of this class of method compared to other more specialized models?
>
> **A2:**
> While our quantitative results may not appear as impressive as previous methods, we note the following advantages of our approach:
>
> - **Performance Trade-off:** Traditional specialized models often undergo fine-tuning on specific datasets for thousands of epochs with carefully optimized hyperparameters to achieve high performance. In contrast, our fine-tuning procedure is lightweight and does not involve extensive hyperparameter tuning across datasets. While this limits our performance, it demonstrates the simplicity and efficiency of our approach.
>
> - **Advantages of Autoregressive Generative Models:** One of the most impressive advantages of our autoregressive generative framework is its task awareness and adaptability. For example, once LVM-Lite is trained, it can seamlessly be deployed for various tasks such as image generation, video frame prediction, and conditional generation without requiring additional training. This flexibility is achieved through an in-context learning procedure, enabling the model to adapt to and infer tasks effectively on the fly.
>
> - **Tokenizer Limitation:** One main issue affecting our quantitative results is the limitation of our tokenizer. As shown in Table 1, the Oracle FID score achievable with our tokenizer is significantly higher than that of previous state-of-the-art methods, indicating room for improvement with better tokenization strategies.
>
> We will include the discussions above in the revised manuscript.
>
> ---
>
> **Q3:** I don’t really see the point of measuring mIoU when the performance is around ~1 or 2 points. I don’t think the comparisons are meaningful, and maybe the qualitative examples are enough to judge the quality.
>
> **A3:**
> We thank Reviewer 2Ntn for their rigor on our image segmentation performance. We evaluated LVM (Bai et al., 2023) on ADE20K and Cityscapes, and obtained comparable mIoU scores of 1.9 and 0.1 respectively with their released checkpoint (see the newly submitted Supplementary Material). The lower quantitative performance on image segmentation originated from LVM. While we focus on improving training efficiently and analyzing LVM behaviors, we will acknowledge this gap in the revised manuscript and leave further performance improvements for future work.
>
> ---
>
> **Q4:** The conclusion that this approach scales with size was already found in [1]. Here the main addition of the paper is to focus on computational efficiency, so it would be interesting to study the sample-efficiency. Given various fixed computational budgets, what performances can you obtain compared to LVM?
>
> **A4:**
> Thanks for the suggestions. We include the comparison with LVM (Bai et al., 2023) using the officially released checkpoints on various benchmarks in the newly submitted Supplementary Material and will report the training compute as well. LVM-Lite attains comparable performance with less compute compared with LVM.
>
> ---
>
> **Q5:** The speedup is only for training time and not inference. Training time is usually less of an issue for practical applications. Have you thought about also improving the architecture to improve inference time?
>
> **A5:**
> We thank Reviewer 2Ntn for raising this important question on inference time speedup. While LVM-Lite is primarily focused on training efficiency improvement, these two-stage techniques are not directly connected to inference time efficiency optimization. We leave it for future work to consider architectural opportunities to speed up LVM inference.
>
> ---

---

> ### Author Response · Authors · 2024-11-29
> **Response to Reviewer 2Ntn (Part 2)**
>
> ### Questions
>
> **Q6:** In Table 3, a better FID correlates to a worse mIoU. Do you have an explanation for that?
>
> **A6:**
> Thanks for the careful review. FID measures the fidelity and diversity of generated images, which prioritizes the quality of the visual output. On the other hand, mIoU evaluates the accuracy of segmentation, which relies heavily on precise pixel-to-pixel correspondence. As shown in our added results, while the generated segmentation masks may appear visually reasonable, the actual class labels and object locations are often misaligned. This indicates that the model struggles with accurately capturing semantic and spatial details, highlighting a gap between visual plausibility and precise object recognition.
>
> ---
>
> **Q7:** Some models have a better FID or IS score compared to the “ground truth rec.” Is it some form of overfitting? How are these metrics reliable compared to just looking at the generated examples?
>
> **A7:**
> The "ground truth rec." represents images reconstructed by the VQGAN model, which itself may introduce artifacts or slight deviations from the true data distribution. If an autoregressive model generates images that appear closer to the statistical distribution of the original dataset than the VQGAN's reconstructions, it could result in better FID or IS scores. This is because these metrics compare the distribution of generated images to the real dataset, not to VQGAN reconstructions.
>
> ---
>
> ### Requested Changes
>
> **Q8:** Provide qualitative, and quantitative if possible, comparisons with LVM on the same prompts to show that the model is of similar quality and that the computational gains are not at the cost of the quality of the model. This is essential for validating the approach, and for me to recommend the paper for acceptance.
>
> **A8:**
> Thanks for the suggestions. We have included the comparisons with LVM in the newly submitted Supplementary Material.

---

### Review · Reviewer_tkS4 · 2024-10-09

**Summary Of Contributions:**

This work builds upon LVM (Bai et al., 2023), an autoregressive Large Vision Model (LVM) that performs various computer vision tasks, such as image and video generation, video prediction, and semantic segmentation, using only in-context examples without requiring any text. During training, LVM uses many sequences of randomly sampled images, which account for around 90% of the training tokens. The authors identify this as a potential issue because these random-image sequences consume a significant portion of the training and are more 'noisy' compared to natural image sequences (e.g., videos).

To speed up the training of LVM, the authors propose a two-stage approach. In the first stage, the model is pre-trained using only single images, which is significantly faster than training with image sequences. In the second stage, the model is fine-tuned using 'Natural Sequences' (e.g., videos), 'Discriminative Sequences' (e.g., sequences of image-annotation pairs), and 'Generative Sequences' (sequences of annotation-image pairs).

The authors evaluate the models on tasks such as video prediction, annotation to image generation (e.g., from segmentation to image), and image segmentation. They demonstrate that using single images during the first training stage significantly speeds up the training compared to using random-image sequences. They also show that using random image sequences during the second fine-tuning stage degrades the performance.

**Audience:**

Yes

**Broader Impact Concerns:**

No concerns.

**Claims And Evidence:**

No

**Requested Changes:**

- The authors need to tone down some of their statements about the strength of their model. For instance, the claims that 'As presented in Table 2 and Figure 4, LVM-Lite achieves comparable performance with specialist models' (Page 6)  and "Our proposed LVM-Lite excels in video prediction and conditional image/video generation, demonstrating comparable performance to specialist models across benchmarks" are somewhat over-stretched. In Table 2 the FID of LVM-Lite on ADE20K-G is NOT comparable to the FID of DP-SIMs. Also, in Table 1 the FVDs score of LVM-lite are NOT comparable with those of MAGVIT. Additionally, stating in the caption of Figure 4 and in the text that LVM-Lite 'can produce high-quality images' is another exaggeration. The generated images in Figure 4 do not look 'realistic' or 'high-quality'.
- It is important to provide some comparisons with the baselines/ablations discussed in the first weakness point, so as to further validate the usefulness of the proposed approach.
- There should be some comparisons with LVM (3rd weakness point).
- (Minor) In Table 6, it would be good to add the training compute in TFLOPs of each training stage (and optionally, the total)
- (Minor) Did the authors explore using more than two training stages with progressive compute requirements, such as a first pre-training stage using only single images, a second pre-training stage using small sequences (e.g., sequences of 2 or 4 examples), and a final training stage using sequences of full length? Perhaps such a multi-stage approach could be exploited to further reduce the training compute.

**Strengths And Weaknesses:**

STRENGTHS:
- Large Vision Models that do not rely on text and can perform various tasks using only in-context examples are an interesting research subject.
- However, they require significant GPU/TPU resources to train them. The authors correctly identify the issue with random-image sequences and propose a simple technique that speeds up the training of such models by 2x to 2.7x without compromising performance.
- This two-stage approach can be very useful for researchers / practioners in the field who want to train LVM models, as it reduces the need for GPU/TPU resources. In other words, this work provides a technique of making research on this field more accessible.

WEAKNESSES:

1. The authors did not validate the usefulness of their first single-image pre-training stage. Specifically, they did not study what the performance would be if the first single-image pre-training stage were skipped and the model were only trained using 'Natural Sequences,' 'Discriminative Sequences,' and 'Generative Sequences' (as in the current second stage). This would further reduce the training compute (since the training compute of the first stage is removed), while making the approach even simpler. Another interesting baseline would be if, in this single-stage training, they also used a small fraction of random-image sequences so that the total training compute is equivalent to the total training compute of their two-stage approach.

2. A major issue with the model is that its image understanding results, particularly its image segmentation quantitative results, are very poor (e.g., 10.3% mIoU on Cityscapes and 2.3% mIoU on ADE20K). Given this, it is hard to claim that the model can perform at all discriminative image understanding tasks. Furthermore, from the image segmentation visualizations in Figures 6 and 7 and the human pose visualizations in Figure 10, it seems that the issue is not a lack of 'pixel-to-pixel' supervision, but rather 'hallucinations': the predictions seem crisp but loosely related to the query image, with fully 'hallucinated' objects.
    - Do the authors use the same tokenizer as in LVM (Bai et al., 2023)? Because the qualitative image segmentation results in LVM seem significantly better than in this manuscript.
    - Also, the image segmentation results for ADE20K do not improve with bigger models (Table 6). Instead, they decrease. So, scaling the model does not seem to be the answer to this issue.
3. Given that the work builds upon LVM (Bai et al., 2023), the lack of comparison with them is an omission.


Miscellaneous:
- The authors claim that "A lower temperature and top-k setting produces more coherent results". However, in the visualizations, the lower temperature values 0.1 and 0.3 do not have more coherent results than the temperature value 0.6.  By the way, it would have been better if the query was always the same when studying the impact of T, Top-K, or prompt num. In other words, having the same query in all the last 3 rows.
- In Table 2 and column ADE20K-G, bold should be the 39.7 instead of the 42.4. Nothing is bold in column City.-G.

---

> ### Author Response · Authors · 2024-11-29
> **Response to Reviewer tkS4 (Part 1)**
>
> We thank Reviewer tkS4 for their constructive comments, and we hope we have addressed their questions adequately, as detailed below.
>
>
> **Q1:** […] what the performance would be if the first single-image pre-training stage were skipped [...]. Another interesting baseline would be if, in this single-stage training, they also used a small fraction of random-image sequences so that the total training compute is equivalent [...]
>
> **A1:**
> Thanks for your suggestions. The significance of single-image datasets has been effectively demonstrated in the LVM paper (Bai et al., 2023). Specifically, Figure 5 illustrates that single-image datasets not only contribute to various downstream tasks, even when trained as random image sequences but also can be integrated with other sequential datasets to further enhance performance.
>
> For the second suggestion to include a fraction of random image sequences during finetuning, we found that the inclusion of random image sequences during fine-tuning is detrimental and degrades performance across all tasks (see Section 3.2 - Sequence Fine-tuning and Table 5 for detailed discussions).
>
>
> ---
>
> **Q2:** A major issue with the model is that its image understanding results [...] are very poor [...] it is hard to claim that the model can perform at all discriminative image understanding tasks. in Figures 6 and 7 [...] the issue is not a lack of 'pixel-to-pixel' supervision, but rather 'hallucinations' [...].
>
> **A2:**
> Thanks for raising the concern about the image understanding results. To better assess LVM-Lite’s segmentation performance, we first reproduced LVM's (Bai et al., 2023) results using their official 7B checkpoint (available at https://huggingface.co/Emma02/LVM_ckpts) (See the newly submitted Supplementary Material). We found that the quantitative segmentation results from LVM were also low despite its visually reasonable qualitative results. Specifically, we evaluated LVM on ADE20K and Cityscapes, obtaining mIoU scores of 1.9 and 0.1 respectively with their released checkpoint. For comparison, our reproduced 300M model achieved a comparable mIoU of 1.7 on ADE20K (see Table 4).
>
> Moreover, Figures 6 and 7 are from image generation tasks instead of semantic segmentation tasks. The quantitative segmentation results mentioned by Reviewer tkS4 were not reported in the LVM paper.
> We do not include a direct qualitative comparison with our segmentation results because we used a different colormap for visualizing segmentation outputs compared to LVM (as shown in Figure 26). This difference makes it challenging to directly compare qualitative results in a meaningful way.
> The new comparison with reproduced LVM (Bai et al., 2023) results using their official checkpoint is available in the newly submitted Supplementary Material. We will tone down the rhetoric to acknowledge the modest performance on image understanding tasks.
>
>
> ---
>
> **Q2a:** Do the authors use the same tokenizer as in LVM (Bai et al., 2023)?
>
> **A2a:**
> Yes, we used exactly the same tokenizer as in LVM (Bai et al., 2023).
>
>
> ---
>
> **Q2b:** Also, the image segmentation results for ADE20K do not improve with bigger models (Table 6).
>
> **A2b:**
> We conjecture that the low image segmentation performance originated from LVM.  Upon closer examination of LVM's results, we found that while the segmentation outputs appear visually similar to the ground truth due to close color matches, the generated classes are often incorrect. This suggests that while the segmentation process itself may be adequate, the model struggles significantly with object recognition, resulting in mismatches between the predicted and actual classes.
>
> While we focus on improving training efficiently and analyzing LVM behaviors, we will acknowledge this gap in discriminative tasks in the revised manuscript and leave further performance improvements for future work, such as using a more effective tokenizer to improve classification and enhance segmentation quality.
>
>
> ---
>
> **Q3:** Given that the work builds upon LVM (Bai et al., 2023), the lack of comparison with them is an omission.
>
> **A3:**
> We apologize for the lack of experimental comparisons with LVM (Bai et al., 2023) in the submitted manuscript, and we include new comparisons using the officially released checkpoint in the newly submitted Supplementary Material. We would like to clarify that, at the time of writing, the official checkpoint from LVM has not been released and thus we had difficulty including it in our first draft.
>
> ---

---

> ### Author Response · Authors · 2024-11-29
> **Response to Reviewer tkS4 (Part 2)**
>
> ### Miscellaneous
>
> **Q4:** (In Figure 6) the lower temperature values 0.1 and 0.3 do not have more coherent results than the temperature value 0.6. it’s better if the query was always the same when studying the impact of T, Top-K, or prompt num.
>
> **A4:**
> We again apologize for the confusion around the results presented in Figure 6. We acknowledge the difficulty of assessing the coherence of generation results with different query inputs. Thus, we will revise Figure 6 to use the same query to maximally demonstrate the impact of T, top-k, and prompt num.
>
> ---
>
> **Q5:** In Table 2 and column ADE20K-G, bold should be the 39.7 instead of the 42.4. Nothing is bold in column City.-G.
>
> **A5:**
> We apologize for the mistake and will rectify this issue in the revised manuscript.
>
> ---
>
> ### Requested Changes
>
> **Q6:** The authors need to tone down some of their statements about the strength of their model. For instance, the claims that 'As presented in Table 2 and Figure 4, LVM-Lite achieves comparable performance with specialist models' (Page 6) and "Our proposed LVM-Lite excels in video prediction and conditional image/video generation, demonstrating comparable performance to specialist models across benchmarks" are somewhat over-stretched. In Table 2 the FID of LVM-Lite on ADE20K-G is NOT comparable to the FID of DP-SIMs. Also, in Table 1 the FVD score of LVM-lite are NOT comparable with those of MAGVIT. Additionally, stating in the caption of Figure 4 and in the text that LVM-Lite 'can produce high-quality images' is another exaggeration. The generated images in Figure 4 do not look 'realistic' or 'high-quality'.
>
> **A6:**
> We thank Reviewer tkS4 for the suggestions. We will tone down our discussions in the revised manuscript to make explicit the gaps between the performance of LVM-Lite and specialist models across benchmarks despite its potential as a vision generalist akin to LLMs, demonstrated from comparisons with LVM (Bai et al., 2023).
>
> ---
>
> **Q7:** It is important to provide some comparisons with the baselines/ablations discussed in the first weakness point, so as to further validate the usefulness of the proposed approach.
>
> **A7:**
> Please refer to our response to Weaknesses #1.
>
>
> ---
>
> **Q8:** There should be some comparisons with LVM (3rd weakness point).
>
> **A8:**
> Please refer to our response to Weaknesses #3 and the newly submitted Supplementary Material.
>
> ---
>
> **Q9:** (Minor) In Table 6, it would be good to add the training compute in TFLOPs of each training stage (and optionally, the total).
>
> **A9:**
> We will add training compute from each training stage to Table 6.
>
>
> ---
>
> **Q10:** (Minor) Did the authors explore using more than two training stages with progressive compute requirements, such as a first pre-training stage using only single images, a second pre-training stage using small sequences (e.g., sequences of 2 or 4 examples), and a final training stage using sequences of full length? Perhaps such a multi-stage approach could be exploited to further reduce the training compute.
>
> **A10:**
> We appreciate the insight provided by Reviewer tkS4 regarding a progressive multi-stage training design for further compute reduction. This is an interesting open question and a natural followup for LVM-Lite with a balanced interplay between efficiency and efficacy. We will include additional discussions for such a multi-stage approach as future directions.

---

### Decision · Action_Editor_H4Lg · 2024-12-09

**Recommendation:** Reject

**Comment:**

This paper proposes LVM-lite, a two-step approach for training autoregressive vision models. Reviewers asked for additional experiments due to missing baselines and ablations, needed to justify the claims made in the paper. Authors were unresponsive towards these requests; the rebuttal deadline was only extended because I proactively reached out to the authors -- only at this point did the authors ask for an extension. This happened twice. The authors finally posted their rebuttal on Nov 29, which I had clearly communicated was the deadline for the reviewers to submit their final recommendations.

While I understand that additional experiments can take time, I find the lack of communication from the authors towards the reviewers really surprising; they could have easily started a discussion during the rebuttal period and only added the requested experiments later. One reviewer had already submitted their recommendation when the authors uploaded their rebuttal and, understandably, decided not to engage with the authors. The other two reviewers mentioned that their concerns had not been fully addressed post rebuttal. I thus recommend rejection.

**Audience:**

Yes, reviewers unanimously agree.

**Claims And Evidence:**

No, see comments below.